# Magnetic Characterization by Scanning Microscopy of Functionalized Iron Oxide Nanoparticles

**DOI:** 10.3390/nano11092197

**Published:** 2021-08-26

**Authors:** Frederico V. Gutierrez, Anna De Falco, Elder Yokoyama, Leonardo A. F. Mendoza, Cleanio Luz-Lima, Geronimo Perez, Renan P. Loreto, Walmir E. Pottker, Felipe A. La Porta, Guillermo Solorzano, Soudabeh Arsalani, Oswaldo Baffa, Jefferson F. D. F. Araujo

**Affiliations:** 1Department of Physics, Pontifical Catholic University of Rio de Janeiro—PUC-Rio, Rua Marques de São Vicente, Rio de Janeiro 22451-900, RJ, Brazil; gutierrez@aluno.puc-rio.br; 2Department of Chemistry, Pontifical Catholic University of Rio de Janeiro—PUC-Rio, Rua Marques de São Vicente, Rio de Janeiro 22451-900, RJ, Brazil; annadefalco9@gmail.com; 3Institute of Geosciences, University of Brasília, Brasília 70910-900, DF, Brazil; eyokoyama@unb.br; 4Department of Electrical Engineering, State University of Rio de Janeiro—UERJ, Rio de Janeiro 20550-900, RJ, Brazil; mendonza@ele.puc-rio.br; 5Department of Physics, Federal University of Piauí—UFPI, Teresina 64049-550, PI, Brazil; cleanio@ufpi.edu.br; 6Department of Mechanical Engineering, Universidade Federal Fluminense—UFF, Rua Passo da Pátrias, n°156, Niteroi 24210-240, RJ, Brazil; geronimoperez@id.uff.br; 7Centro Brasileiro de Pesquisas Físicas/MCTI, CBPF, Rio de Janeiro 22290-180, RJ, Brazil; renan.loreto@gmail.com; 8Federal Technological University of Paraná, UTFPR, Avenida dos Pioneiros 3131, Londrina 86036-370, PR, Brazil; walmir@utfpr.edu.br (W.E.P.); felipelaporta@utfpr.edu.br (F.A.L.P.); 9Department of Chemical and Materials Engineering, Pontifical Catholic University of Rio de Janeiro—PUC-Rio, R. Marquês de São Vicente, 225, Gávea, Rio de Janeiro 22430-060, RJ, Brazil; guilsol@puc-rio.br; 10Physikalisch-Technische Bundesanstalt, Abbestrasse 2-12, D-10587 Berlin, Germany; sudabeh.arsalani@gmail.com; 11Department of Physics, FFCLRP, University of Sao Paulo, Av. Bandeirantes 3900, Ribeirão Preto 14040-91, SP, Brazil; baffa@usp.br

**Keywords:** magnetic nanoparticles, co-precipitation, Pluronic F-127, scanning magnetic microscope

## Abstract

This study aimed to systematically understand the magnetic properties of magnetite (Fe_3_O_4_) nanoparticles functionalized with different Pluronic F-127 surfactant concentrations (Fe_3_O_4_@Pluronic F-127) obtained by using an improved magnetic characterization method based on three-dimensional magnetic maps generated by scanning magnetic microscopy. Additionally, these Fe_3_O_4_ and Fe_3_O_4_@Pluronic F-127 nanoparticles, as promising systems for biomedical applications, were prepared by a wet chemical reaction. The magnetization curve was obtained through these three-dimensional maps, confirming that both Fe_3_O_4_ and Fe_3_O_4_@Pluronic F-127 nanoparticles have a superparamagnetic behavior. The as-prepared samples, stored at approximately 20 °C, showed no change in the magnetization curve even months after their generation, resulting in no nanoparticles free from oxidation, as Raman measurements have confirmed. Furthermore, by applying this magnetic technique, it was possible to estimate that the nanoparticles’ magnetic core diameter was about 5 nm. Our results were confirmed by comparison with other techniques, namely as transmission electron microscopy imaging and diffraction together with Raman spectroscopy. Finally, these results, in addition to validating scanning magnetic microscopy, also highlight its potential for a detailed magnetic characterization of nanoparticles.

## 1. Introduction

A spectrum of current studies in nanotechnology highlights the rapid advance that has been taking place in this area of science, especially in medical applications. This combination of nanotechnology, specifically the use of nanostructured materials, with medicine has shown that it is possible to omit the level of abstraction, making it a reality that could benefit many patients [1]. We can cite some examples such as the use of substrates of nanomaterials that help stem cell growth [2], nanofibers in dental applications, medical implants, tissue engineering and many other applications [3]. In any study where we relate these two areas, we need to use materials with high biocompatibility or combine them to achieve that. In this way, research on this topic focuses on developing new biomaterial assembly techniques to functionalize inorganic nanomaterials, in specific magnetic nanoparticles [4,5].

The study of functionalized magnetic nanoparticles (MNPs) has shown tremendous technological advantageous features for biomedical applications [6,7,8], such as MRI contrast [9,10] and drug delivery systems [11], to increase efficiency by releasing a specific drug in a target tissue, without possible damage to healthy tissues [5,12,13]. In the presence of a magnetic field, MNPs can be driven to a specific location, increasing the drug concentration [14,15]. However, in this case, it is essential to study the magnetic response of MNPs, expecting that such materials have a superparamagnetic behavior at near room temperature, which is necessary to avoid particle agglomeration due to a remaining magnetization [6,15,16,17].

The physical properties of MNPs are generally correlated with their composition, morphology, size, spatial distribution, and also with their degree of crystallinity. As we have known, the MNPs’ core must preferably be around 10–50 nm, as promising candidates for diverse biomedical applications as well as contribute to prevent intravenous clogging while maintaining the superparamagnetic behavior [13,17]. Particularly, the agglomeration effect of MNPs (due to wide size distribution) under biological conditions, in principle, might significantly be reduced after functionalization with surfactants [13,17,18]. Regarding synthesis, it is well-known that a significant variety of MNPs has been easily produced with high yields, purity, and also at a relatively low cost through the sol-gel and co-precipitation methods [13,17,19,20].

As we have known, the products from precipitation reactions are generally soluble species formed under high supersaturation conditions, which significantly affect the product size and morphology, and their physical properties [21]. Raman spectroscopy, one of the characterization techniques that has been widely used to distinguish phases and/or polymorphism [22,23,24] by present distinct Raman signatures, either by difference in crystal structure or by change in oxidation state, but also commonly used in characterizing different morphologies [23,25,26], oxidation states [27,28], phase transitions [29,30], and due alteration doping [28,30] is Raman spectroscopy. Faria et al. [22] investigated seven phases of iron oxides by Raman spectroscopy, including magnetite (Fe_3_O_4_) and hematite (α-Fe_2_O_3_), and showed that the Raman spectra are distinct for each phase, making this technique suitable to determine the iron oxide phases. Furthermore, a phase transition has been reported from maghemite (γ-Fe_2_O_3_) and magnetite Fe_3_O_4_ to hematite α-Fe_2_O_3_ by Raman spectroscopy varying the laser power, respectively [31,32].

In recent years, the scanning magnetic microscope (SMM) has played a fundamental role in the characterization of magnetic materials, updated in the characterization of samples of several materials, such as rocks, steels, and nanostructures [33,34,35]. In the present work, we report a magnetization method improvement, where the maps obtained by a home-built SMM are three-dimensional (3D) instead of the in-line technique obtained through maps in two-dimensional (2D) mode, as previously developed by Araujo et al. [33]. Parameters extracted from the SMM data can be used to build a MxH hysteresis curve with the respective magnetization values for each applied field. Those values were compared to vibrating sample magnetometer (VSM) measurements, at room temperature, where the sample oscillates close to a detection coil, and an induced voltage appears, corresponding to the magnetization of the sample and a superconducting magnet generates the external magnetic field [36]. As a result, these Fe_3_O_4_ and Fe_3_O_4_@Pluronic F-127 nanoparticles (NPs) exhibit a superparamagnetic behavior at room temperature. Transmission electron microscopy (TEM) and Raman results also show that the synthetized samples are composed mostly of magnetite, some of them containing small amounts of hematite. These results are crucial for most of the above-mentioned applications to avoid the agglomerations of MNPs, in addition to the reduction in oxidation.

## 2. Experimental

The Fe_3_O_4_ and Fe_3_O_4_@Pluronic NPs were produced using the co-precipitation method [9,13,16,17,18] using Fe^3+^ and Fe^2+^ ions, in a 2:1 ratio. This synthesis consists of adding a homogeneous mixture of iron salts to a basic solution containing NH_4_OH, at a temperature of approximately 80 °C to exceed the desired salt solubility product for precipitation [13,16,17,18,19]. The reagents used in the synthesis consisted of ferric chloride hexahydrate (FeCl_3_·6H_2_O, 270.29 g mol^−1^), iron sulphate (II) heptahydrate, (FeSO_4_·7H_2_O, 278.01 g mol^−1^), ammonium hydroxide (NH_4_OH, 35.04 g mol^−1^), and hydrochloric acid (HCl, 36.46 g mol^−1^). All reagents were of analytical grade and were produced by Vetec Química Fina LTDA, Rio de Janeiro, Brazil.

The following production procedures were used, as illustrated in Figure 1: (1) Two solutions were prepared for the iron salt mixture, one dissolving 0.02516 mol of FeCl_3_·6H_2_O in 25 mL of distilled water and another dissolving 0.01421 mol of FeSO_4_·7H_2_O in 10 mL of HCl (5.49 mol L^−1^); (2) the iron salt solutions were mixed in a ratio of 4 mL Fe^3+^ to 1 mL Fe^2+^, in agreement with the ratio of 2:1; (3) an aqueous solution of NH_4_OH (1.30 mol L^−1^) was preheated to 80 °C in a separate container on a heating plate for 10 min; (4) the obtained salt mixture was added to the basic solution of ammonium hydroxide (28% P.A.) under manual vigorous stirring with a glass rod; (5) a black precipitate was formed, indicating the formation of NPs; (6) the MNPs were kept in the ultrasound bath for one hour to prevent agglomeration; (7) subsequently, the MNPs were washed three times with distilled water (distiller NT 422, NovaTecnica, Brasil) and the help of a permanent magnet to hold the nanoparticles. The washing procedure was repeated several times to neutralize the solution pH.

The procedures for coating Fe_3_O_4_ samples with Pluronic surfactants (Figure 2) were performed, according to the procedure, as follows: (1) solutions at different concentrations of Pluronic F-127 were prepared (Table 1); (2) a 10 mL aliquot was taken from each solution and added to the aqueous NH_4_OH solution (1.28 mol L^−1^); (3) the basic solution was heated to 80 °C on a heating plate for 10 min; (4) the iron salt solution was added to the Pluronic F-127 (Sigma-Aldrich, St. Louis, MI, USA) solution under vigorous stirring, using a nonmagnetic spatula; (5) a black precipitate product was immediately observed, indicating the formation of NPs; (6) the NPs were kept in an ultrasound bath for one hour to prevent agglomeration; (7) the same procedure used for washing MNPs was performed for the samples coated with Pluronic F-127. After these procedures, we could observe the NPs (Figure 3), that were stored at approximately 20 °C.

## 3. Nanoparticles Characterization

Uncoated and functionalized MNPs were characterized using Raman spectroscopy, transmission electron microscopy (TEM) scanning magnetic microscopy (SMM) and vibrating sample magnetometer (VSM), as described below.

### 3.1. Raman Spectroscopy

Raman spectra were obtained from the accumulation of 10 spectra of 20 s each, using a Senterra Bruker micro-Raman spectrometer equipped with a CCD system and adjusted to a resolution of 4 cm^−1^. A 785 nm solid-state laser was used as excitation source, with 20 and 50 mW power. The spectrometer uses an Olympus BX50 microscope with 50× objective to focus the laser beam onto the sample surface. To avoid local heating effects, the laser was set to 20 mW and 50 mW was used to examine local heating effects.

Both the magnetite [31] and maghemite [32], a phase transition to hematite products were was investigated by varying the laser power. The spectra shown in Figure 4 were obtained with a power of 20 mW at 0 (black lines) and 6 (red lines) months after synthesis. By using a 20 mW power, we can guarantee the early phase spectrum of materials without any interference from the laser power. The bands observed in the spectra, shown in Figure 4, at 293, 345, 500, and 685 cm^−1^ are characteristic of the magnetite phase [22,31,37,38], although hematite phase bands (222 and 400 cm^−1^) have been observed in some spectra [22,39], featuring a phase mixture. Furthermore, the high similarity between the spectra taken at 0 and 6 months (See Figure 4a), for each sample, is a strong indication of the particles stability and assures us that the sample has not been showing any oxidation process over time. The position and FWHM were obtained through the deconvolution of Raman spectra using the Origin 6.0 software, the result for NP0 e NP5 with 0 months are shown in Figure 4b,c.

In addition, Chandramohan et al. [40]—considering that in nano-sized systems, due to lack of long range order, scattering with q≠0 is allowed, which results at the appearance, widening and displacement of the peak position in the Raman spectra [40]—proposed an relation for estimating size of the magnetite particles, Equation (1) based on Raman spectra (modes A_1g(1)_ (685 cm^−1^) and TEM micrographs for the ferrite particles synthesized by the double microemulsion and polymer combustion methods.
(1)$$\Delta \omega =\omega \left(L\right)-\omega_{0}=A{\left(\frac{a}{L}\right)}^{\gamma }\;$$
where Δω is the variation between the frequency of Raman peak ($\omega \left(L\right)$) with particle size L and the frequency at the Brillouin zone center ($\omega_{0}$). a is the lattice constant of the crystal and, A and γ are the fit parameters that describe the phonon confinement in nano-crystallites. Hence, we used the inverse function of the relation proposed by Chandramohan et al., Equation (1), to determine an expression for estimating size of the particles $\left(L\right)$ in function of Δω, in this case, Δω is full width at half maximum (FWHM), as follows:(2)$$L=a{\left(\frac{A}{\Delta \omega }\right)}^{\frac{1}{\gamma }}\;$$
with a=8.38 Å, $A=104\;{\mathrm{cm}}^{-1}$, and γ=0.8 as constants, and obtained by Chandramohan et al. The estimated value of particle sizes based on the FWHM of the 685 cm^−1^ peak are shown in Table 2.

The Raman spectra obtained with a power of 50 mW for the same samples revealed the presence of Raman peaks at 221, 284, 400, 488, and 607 cm^−1^ (see Figure 5). These values are typical of α-Fe_2_O_3_ and are in accordance with the literature [22,39,41,42,43]. Peaks at 671 and 710 cm^−1^ were also observed and according to the literature, the appearance of these peaks in the hematite phase may be related to the nanometric size of the particles [41,42,44]; such activation shows that the disorder of the nanomaterial surface induces the phonon scattering symmetry breaking [45], which explains why this peak is not always observed in the spectra. In addition, the 400 and 607 cm^−1^ vibrational modes show a gradual softening with increasing percentage of Pluronic surfactant, indicating the presence of Pluronic bound to the MNPs.

### 3.2. Transmission Electron Microscopy

Powder samples were diluted in an aqueous medium and deposited on a 400-mesh carbon-coated copper grid. After the drying process, the samples were analyzed in a TEM (Jeol 2010 instrument), operating at 200 kV.

The six samples analyzed in this set of TEM micrographs (Figure 6) have similar shapes, sizes, and modes of aggregation of nanoparticles. Spheroidal nanoparticles (around 5–10 nm) were predominant among the samples; however, a small amount of larger faceted shapes (30–40 nm) were observed (Figure 6).

The high-resolution micrographs of the six samples (Figure 7), show the atomic planes of spheroidal nanoparticles with size of around 5 nm. Figure 8 shows a high-resolution image of the NP4 sample, highlighting the atomic plane corresponded to the (220) family, with 0.29 nm spacing, of the magnetite spinel structure indexed with ICDD PDF: 01-075-0449, observed along the zone axis (111).

The electron diffraction (ED) pattern shown in Figure 9 indicates a typical structure of magnetite with spinel structure and a very little fraction of hematite, as shown in Figure 9e. These results show that the samples consist of an inverse spinel structure which corresponds to magnetite. The impurity phase of hematite has been found in the NP4. The mean particle size and distribution were evaluated by measuring the largest internal dimension of at a thousand particles of each sample. The six samples displayed a similar size distribution. They had an average size on the order of 5 nm (spheroidal morphologies), with the presence of a small number of larger particles of size around 30–40 nm (faceted shapes). The statistical analysis (of population of measured particles (N), mean size (nm), standard deviation (SD), standard error (SE), minimum size (Min), maximum size (Max), and range) were obtained using the Origin 6.0 software (Figure 10 and Table 3).

### 3.3. Magnetometer and Scanning Magnetic Microscope

The magnetic maps were generated using a home-built SMM, with a reading system containing Hall effect sensors connected to a reading system [33,34]. The SMM technique consists of an improved method to produce magnetic maps for each magnetic field applied to the sample. In this case, the scan is made by moving the sample in a XY stage inside the magnetic field, applied generated by an electromagnet. A drawing sketch of the microscope assembly can be found on references [33,34]. The magnetic field induced from the sample is detected by two Hall effect sensors, connected to standard amplifiers, filters and A/D converters to optimize the detection of the sample field. The reading system is fixed on a printed circuit board and the sample, placed in a sample holder, moves in XY space through a system composed of two stepper motors. The applied magnetic field can be set up to 0.5 T and has a magnetic moment sensitivity of approximately 10^−11^ Am^2^ [33,34].

Approximately tens of micrograms of MNPs were placed into a cylindrical cavity measuring 400 µm in diameter and 400 µm deep (See Figure 11a). In the previous method, the magnetization curve was obtained by a technique that required several points along a two-dimensional map to obtain the magnetic moment. This method was improved by obtaining the magnetic moment using only two points of the 2D map, one in the region of minimal induced field strength of the sample and another the second point exactly in the region of maximum induced magnetic field strength of the samples. The minimal field measurement must be performed in the sample holder (away from the cylindrical cavity), as shown in Figure 11b. These two regions are necessary to obtain the sample magnetic moment using a theoretical model that considers the sample holder cylindrical shape [33,34,46].

For each magnetic field applied by the electromagnet, a magnetic scanning map of the induced field from the sample, from positive values of external fields to negative fields (i.e., 510 mT to −510 mT and returning to 510 mT), were produced and a hysteresis cycle was thus completed (Figure 12). From these 3D graphs, the maximum and minimum regions necessary to obtain the sample magnetic moment could be precisely obtained for each applied magnetic field, from where we can build a hysteresis curve. Despite the limitation of the magnetic field strength applied to the sample NP0, we can see that the hysteresis curve (Figure 13) indicates a superparamagnetic behavior material, as there is neither coercivity nor remanence [33,34].

This same NP0 sample was measured in a commercial Quantum Design PPMS (model VersaLab), where the measurements were made in vibrating sample magnetometer (VSM) mode. The comparison between the two measurements is shown in Figure 13. The curve composed of blue circles resulted from the measurement obtained by SMM at PUC-Rio and the continuous curve, in red, is the measurement made by VSM, showing a good agreement with each other. The average diameter of the MNPs could also be estimated. Although there is a size distribution, as observed in the TEM images, for this estimate, all particles were considered as having the same size, disregarding any interaction. In this case, the Langevin equation was used, making a first-order approximation to low magnetic fields [47,48]. That is:(3)$$x=\left[\frac{{µ}_{0}M_{s}^{2}V}{3K_{B}T}\right]$$
where x is the magnetic susceptibility per volume, ${µ}_{0}$ is the magnetic permeability of vacuum, *M_s_* is the saturation magnetization, *V* is the volume to be estimated, $K_{B}$ the Boltzmann constant, and *T* is the temperature (all measurements were made at room temperature, 293 K). Equation (3) can be rewritten as a function of the diameter (*D*), considering the particles are spherical,
(4)$$D={\left[\frac{18\chi K_{B}T}{{µ}_{0}M_{s}^{2}\pi }\right]}^{1/3}$$

Thus, the average diameter of NP0 could be estimated. The following values were used to estimate the MNPs diameter, based on measurements made: x = 2.7, obtained by using a linear low field approximation of the MxH initial curve, (taken from blue circles on Figure 13), and using magnetite density ρ = 5.197 × 10^3^ Kg m^−3^ [49] and *M_s_* = 754 kA m^−1^ (this value was also estimated by extrapolating the magnetic magnetization curve M in A/m as a function of 1/H magnetic field applied in A m^−1^), we obtain a diameter of 4.4 nm, according to Equation (4). The values for estimating the average diameter, the standard deviation σ for the two magnetic measurements, and the estimate using the other characterization techniques are summarized in Table 4. This diameter estimation is based on the magnetic properties of the material, considering only the nanoparticle magnetic core, subtracting the PL-127 shell.

Although the estimated diameters on Table 4 are in the same order of magnitude, the differences among them are due to some factors: (i) the Raman measurements were taken at low powers, since phase changes occur for high laser powers, and this leads to a low-resolution spectra; (ii) TEM measurements shows a wide distribution of particle size, ranging from 1.5 to 39 nm; and (iii) the diameter estimation made by using Equation (4) is carried out by assuming that all particles are spherical and have the same size. Additionally, this calculation does not consider the Pluronic coating of MNPs, which might affect the TEM and Raman calculations.

This same method was also applied to samples NP1, NP2, NP3, NP4, and NP5. The same superparamagnetic behavior of these materials was observed for all Pluronic coated NPs (Figure 14, showing the NP2 maps), with no remanence and coercivity as well, shown in Figure 15a (NP2) and Figure 15b (NP5).

The values for magnetization at the limit of each magnetic field applied (see Figure 16a) are shown in Table 5 and summarized in Figure 16b. These results suggest that, after coating, there was a change in the magnetization curve in most samples. An important result of this magnetic characterization is that the Pluronic coating is performed during the mixing of iron salts and not after the production of NPs [5,17]. The results of the magnetic characterization curves, typical of these processes, are represented by a pronounced decrease in magnetization, as can be seen in Figure 16a, where we can also note that, even applying 2.0 T and after coating, all samples preserve the superparamagnetic behavior at room temperature.

## 4. Conclusions

In summary, we successfully synthesized magnetite nanoparticles using the co-precipitation method, coating them with a Pluronic F-127 surfactant, during the synthesis, in the mixture of iron salts at different concentrations. The structural properties of the samples were characterized by Raman spectroscopy and transmission electron microscopy. From the Raman results, we verified that the characteristic bands of magnetite phase of Fe_3_O_4_ were observed in the spectra, at 195, 293, 545, and 685 cm^−1^. However, in some samples, we also observed some bands from the hematite phase, also confirmed by ED patterns results of the diffraction patterns acquired with the transmission electron microscope. The as-prepared samples, stored at approximately 20 °C, showed no change in the Raman spectrum, even after six months, indicating no oxidation of the material. In some samples, the hematite appeared during the synthesis, and it was observed even in the 0 months Raman spectrum, indicating that there was no oxidation or phase change throughout time. Using the Raman spectrum and TEM images, it was possible to estimate the diameter of the synthetized nanoparticles, with values of 7.8 nm and 5.4 nm, respectively.

The magnetic characterization of the coated nanoparticles was carried out by using a commercial vibrating sample magnetometer and a home-built scanning magnetic microscope. The novel SMM technique uses a set of Hall sensors for measuring three-dimensional magnetic maps of a sample. By this measurement, the values of the magnetic moment of a sample, in a cylindrical cavity, for an applied magnetic field, can be extracted and it is possible to build a MxH hysteresis loop. The magnetization curves from the SMM data are compared to a VSM measurement, with very good agreement. Both SMM and VSM measurements shows that all samples exhibit a superparamagnetic behavior, suitable for biomedical applications. Furthermore, by the magnetic measurements it was also possible to estimate the nanoparticles’ magnetic core diameter about 4.4 nm, which is consistent with TEM and Raman measurements.

## Figures and Tables

**Figure 1 nanomaterials-11-02197-f001:**
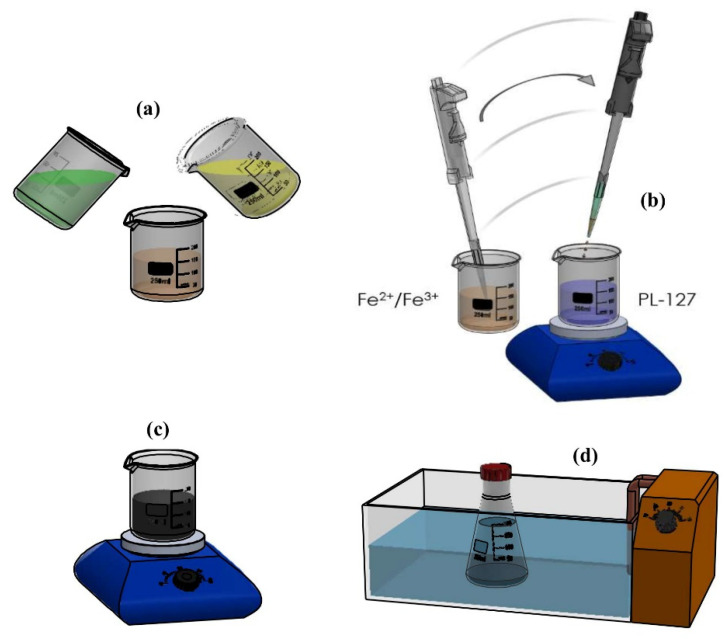
Synthesis of iron oxide NPs steps. (**a**) Formation of iron salt solution from the Fe^2+^ and Fe^3+^ mixture, (**b**) addition of the iron salt mixture to the basic solution of NH_4_OH, (**c**) Becker with MNP precipitate, and (**d**) MNP sample in an ultrasound bath.

**Figure 2 nanomaterials-11-02197-f002:**
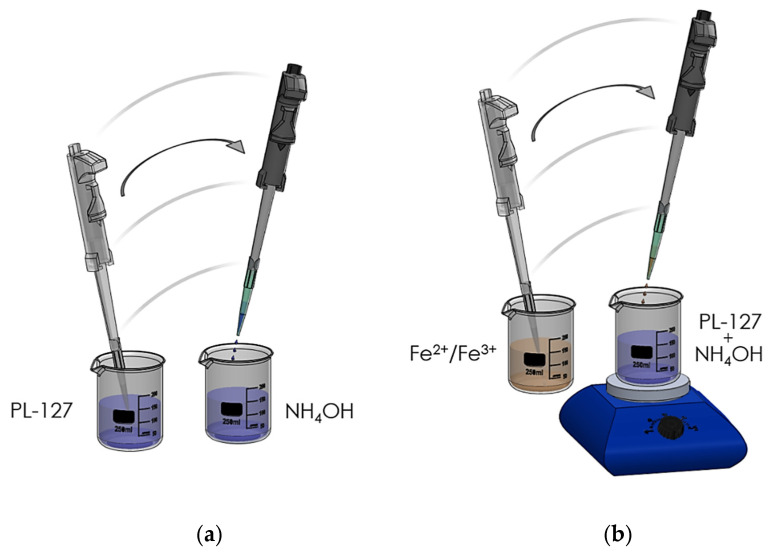
Synthesis of PL-127-coated MNPs. (**a**) Addition of Pluronic F-127 to the basic NH_4_OH solution, (**b**) addition of iron salt mixture to the basic NH_4_OH solution.

**Figure 3 nanomaterials-11-02197-f003:**
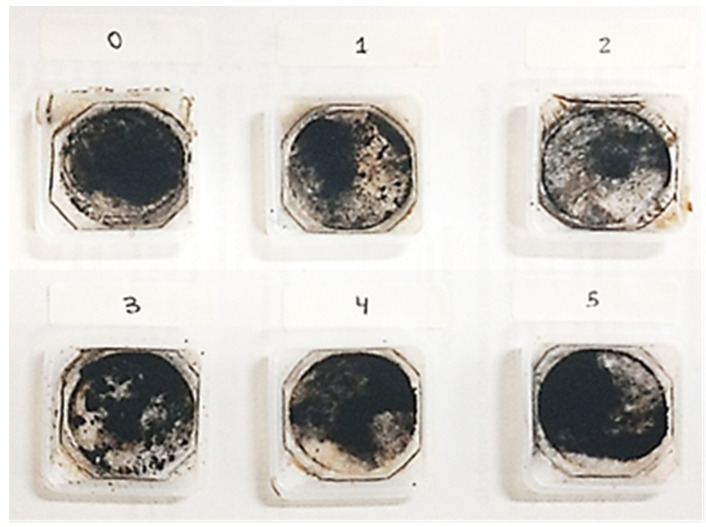
Samples of iron oxide NPs after drying. Pure Fe_3_O_4_ NP sample (NP0), functionalized with progressive amounts of Pluronic F-127 (NP1, NP2, NP3, NP4, and NP5). For more information, see Table 1.

**Figure 4 nanomaterials-11-02197-f004:**
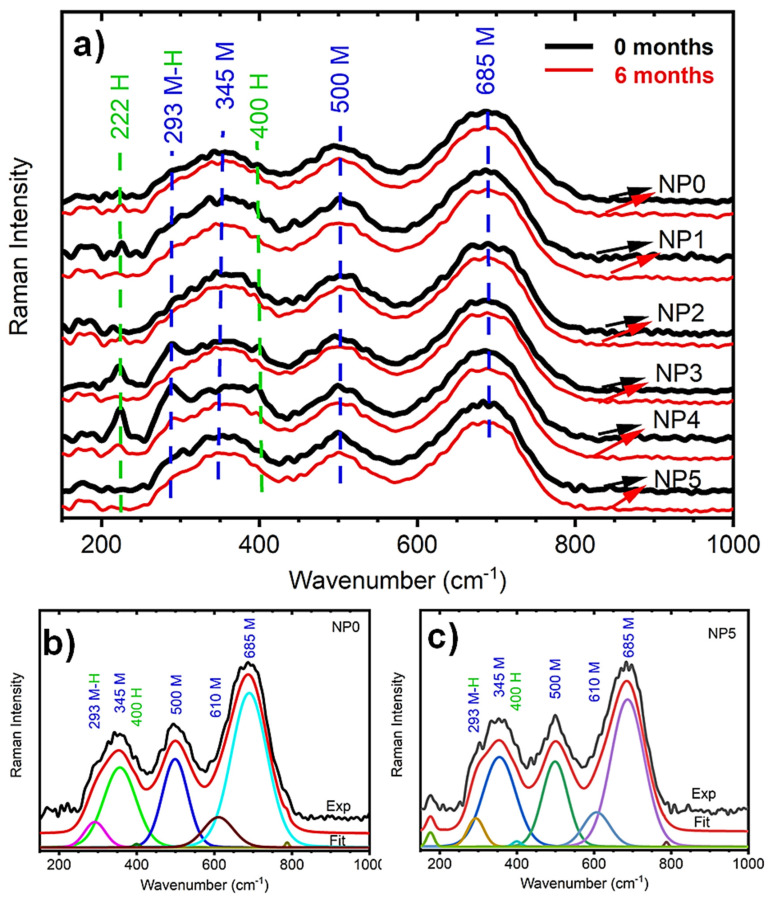
Raman measurements (**a**) taken at 0 and 6 months after synthesis for each sample obtained with a laser power of 20 mW, in which the spectral characteristic bands are predominant in the magnetite phase (**a**). Deconvolution of the Raman peaks for samples NP0 (**b**) and NP5 (**c**), in which we extracted the data for Table 5.

**Figure 5 nanomaterials-11-02197-f005:**
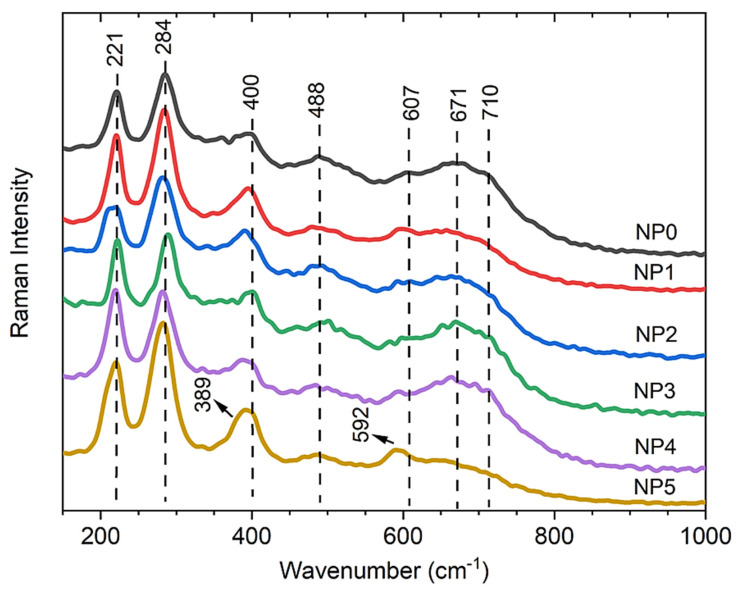
Raman spectra obtained with the power of 50 mW, in which the spectrum characteristic bands are predominant in the hematite phase.

**Figure 6 nanomaterials-11-02197-f006:**
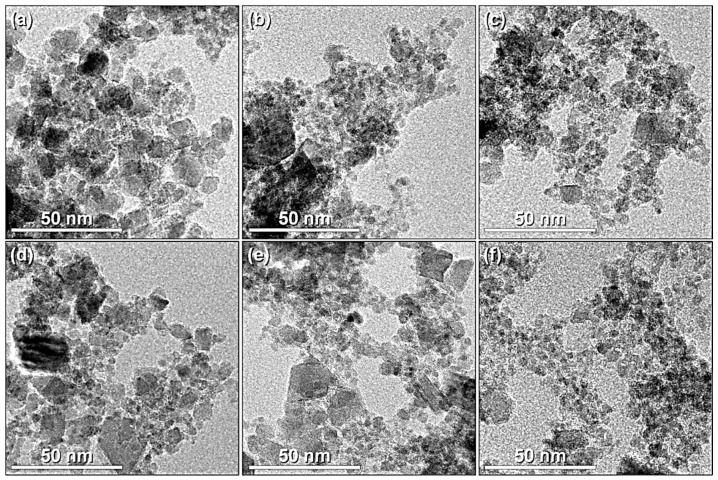
TEM images of the samples: (**a**) NP0, (**b**) NP1, (**c**) NP2, (**d**) NP3, (**e**) NP4, and (**f**) NP5.

**Figure 7 nanomaterials-11-02197-f007:**
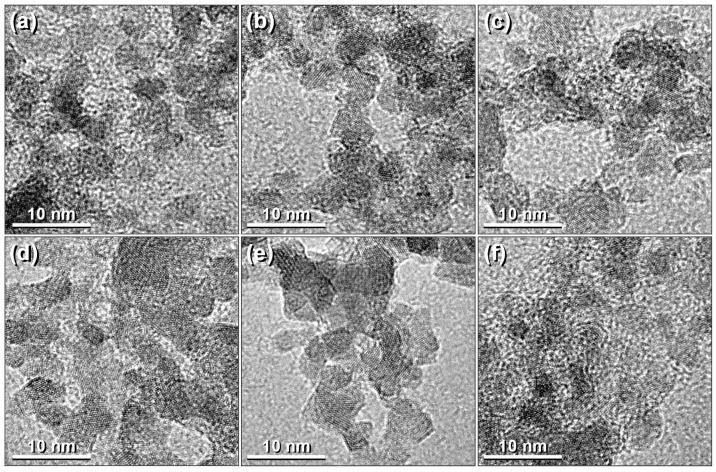
HRTEM images of the samples: (**a**) NP0, (**b**) NP1, (**c**) NP2, (**d**) NP3, (**e**) NP4, and (**f**) NP5.

**Figure 8 nanomaterials-11-02197-f008:**
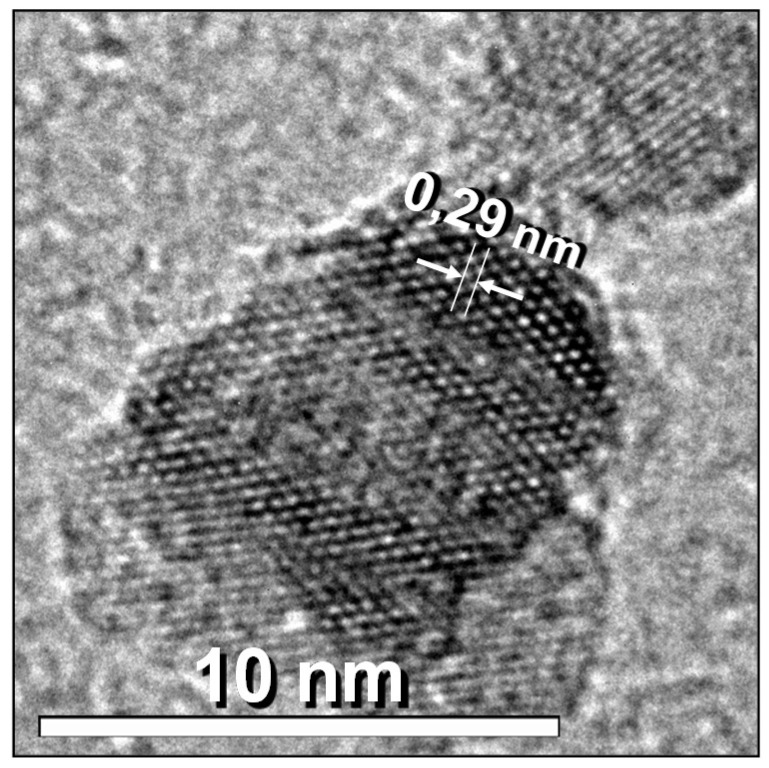
HRTEM image of the sample NP4: detail of atomic columns of 0.29 nm spacing corresponding to (220) magnetite planes.

**Figure 9 nanomaterials-11-02197-f009:**
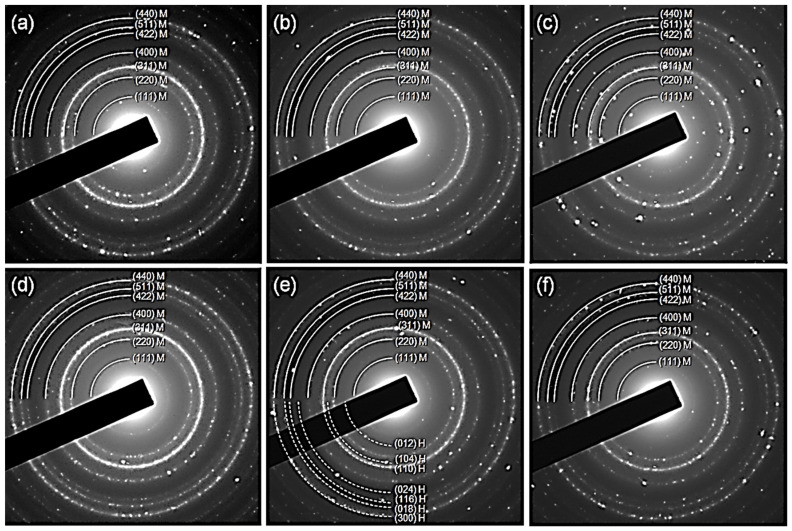
Electron diffraction (ED) patterns of the samples (**a**) NP0, (**b**) NP1, (**c**) NP2, (**d**) NP3, (**e**) NP4, (**f**) NP5. M: magnetite, H: hematite.

**Figure 10 nanomaterials-11-02197-f010:**
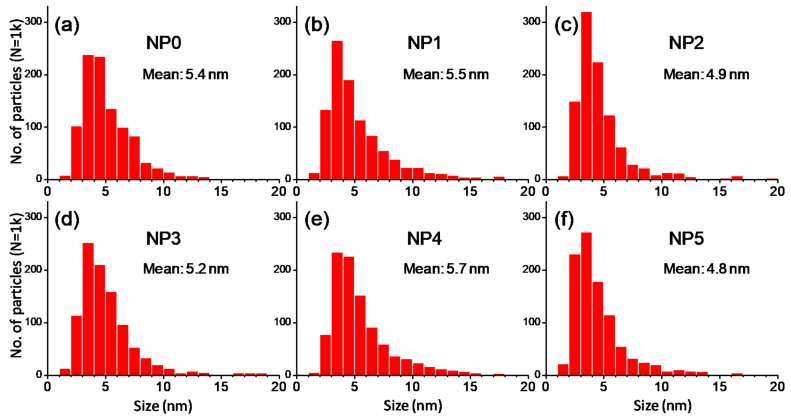
Histograms of particle size of the samples (**a**) NP0, (**b**) NP1, (**c**) NP2, (**d**) NP3, (**e**) NP4, (**f**) NP5. Population: 1000 particles per sample.

**Figure 11 nanomaterials-11-02197-f011:**
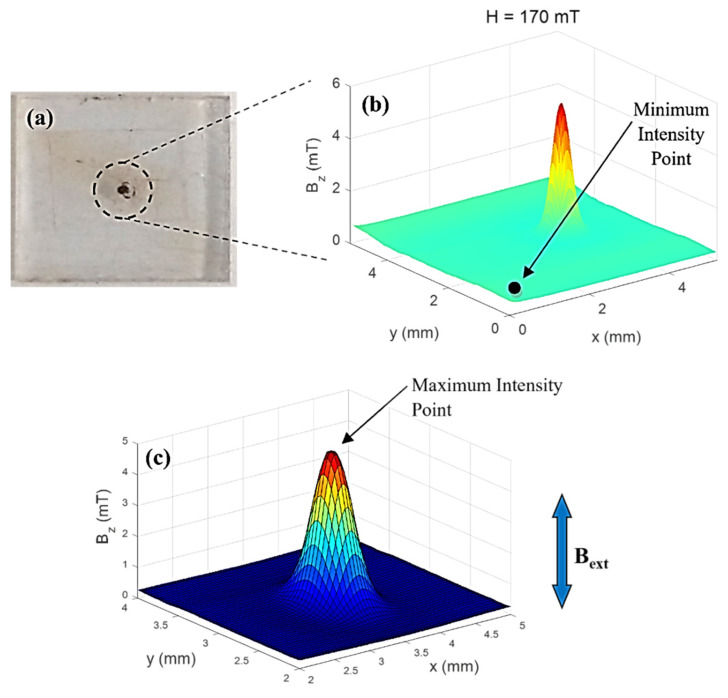
SMM measurements: (**a**) Sample holder made of acrylic; in the center, we can see the cylindrical cavity where the NP0 were placed. (**b**) Map Magnetic in 3D, with an applied magnetic field of 0.17 T. (**c**) Map Magnetic in 3D with grid where it is possible to verify the possibility of maximum intensity. The direction of external magnetic field is displayed.

**Figure 12 nanomaterials-11-02197-f012:**
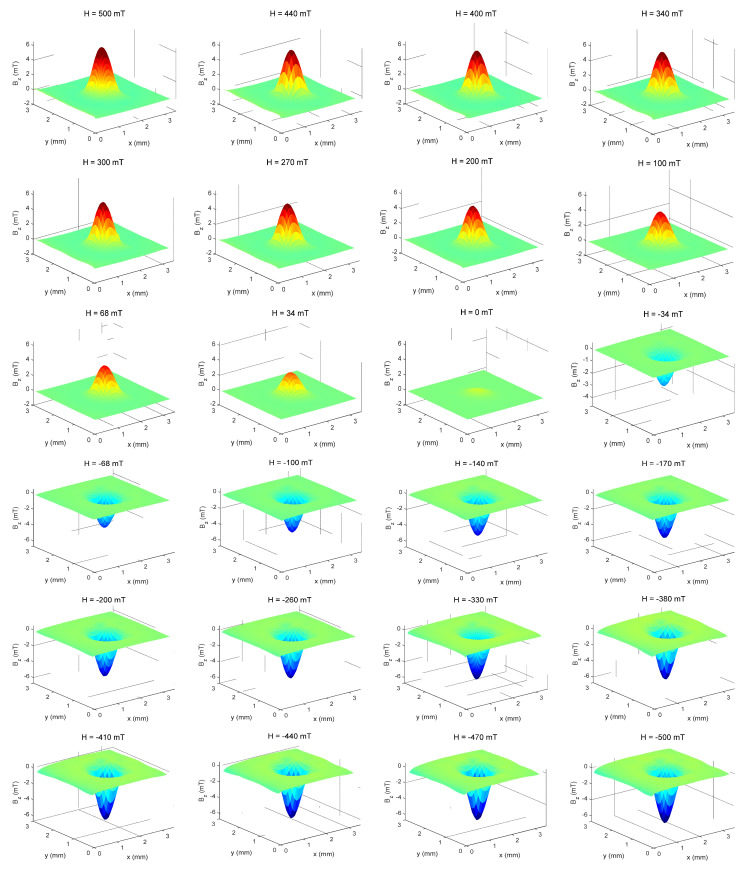
Magnetic maps of the sample NP0 in 3D, obtained directly from the maps in XY space from 500 mT to −500 mT.

**Figure 13 nanomaterials-11-02197-f013:**
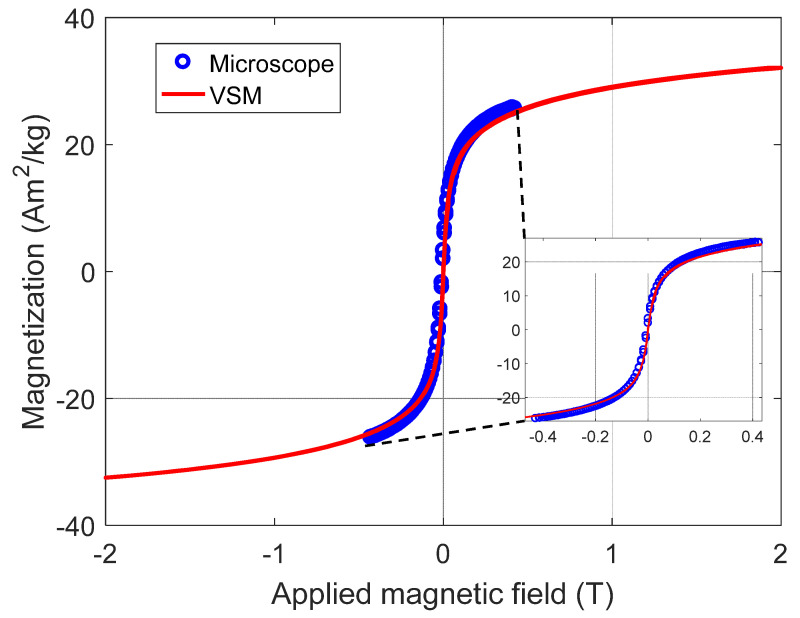
Graph representing the measurements of the sample NP0. The curve composed of blue circles is taken from data obtained from the measurement made in the magnetic microscope of PUC-Rio and the continuous curve in red color is the measurement made in the commercial magnetometer VSM.

**Figure 14 nanomaterials-11-02197-f014:**
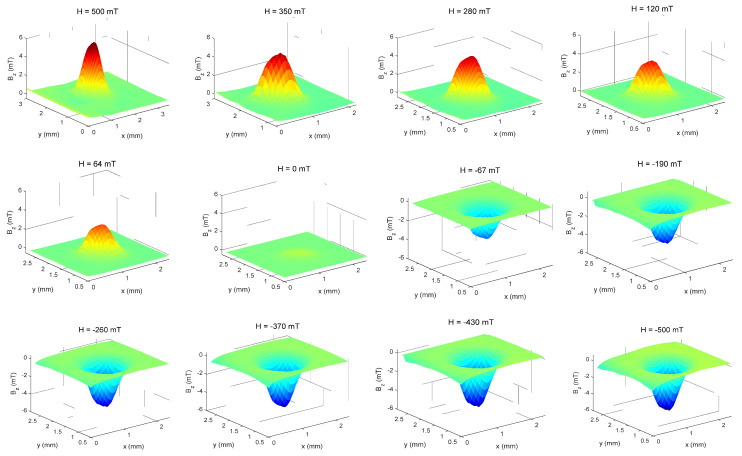
Magnetic maps of the sample NP2 in 3D, obtained directly from the maps in XY space.

**Figure 15 nanomaterials-11-02197-f015:**
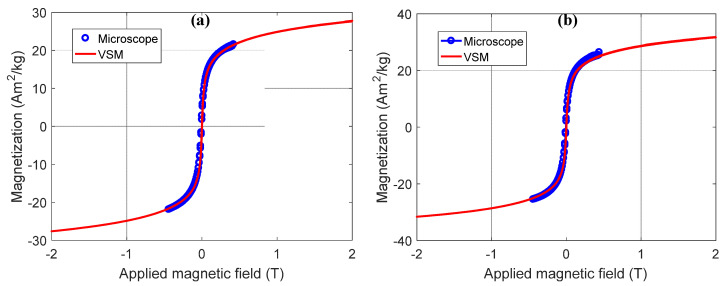
Magnetic hysteresis loops for two samples, measured by VSM and extracted from SMM data. (**a**) Graph representing the measurements of sample NP2. The curve composed of blue circles is the measurement made in the magnetic microscope of PUC-Rio and the continuous curve in red color is the measurement made in the commercial magnetometer of the CBPF. (**b**) Graph representing the measurements of sample NP5. The curve composed of blue circles is the measurement made in the magnetic microscope of PUC-Rio and the continuous curve in red color is the measurement made by VSM.

**Figure 16 nanomaterials-11-02197-f016:**
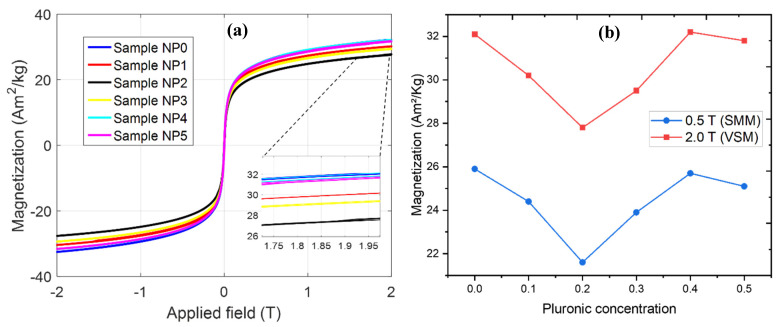
Magnetization measurements (**a**) of the six samples by VSM. The inset shows the saturation region, close to a magnetic field of 2 T. (**b**) Values of saturation magnetization versus Pluronic consent.

**Table 1 nanomaterials-11-02197-t001:** Aqueous solutions of NH_4_OH at different concentrations of Pluronic F-127.

Sample Code	Pluronic F-127 Concentration	Pluronic F-127 (mL)	H_2_O (mL)	Aqueous Solution NH_4_OH (mL)
NP5	0.500	10	0	50
NP4	0.400	8	2	50
NP3	0.300	6	4	50
NP2	0.200	4	6	50
NP1	0.100	2	8	50
NP0	0.000	0	10	50

**Table 2 nanomaterials-11-02197-t002:** Particle size statistics from peak 685 cm^−1^. The sizes were calculated using Equation (2).

Sample	Position (cm^−1^)	FWHM (cm^−1^)	Size (nm)	Standard Error (nm)
NP0	690.1	109.3	7.8	±0.3
NP1	687.3	102.8	8.5	
NP2	689.9	108.9	7.9	
NP3	691.6	106.0	8.1	
NP4	691.0	100.8	8.7	
NP5	686.9	102.3	8.5	

**Table 3 nanomaterials-11-02197-t003:** Statistical data of particle size: population of measured particles (N), mean size (nm), standard deviation (SD), standard error (SE), minimum size (Min), maximum size (Max), and range.

Sample	N	Mean (nm)	SD	SE	Min (nm)	Max (nm)	Range
NP0	1000	5.37	3.17	0.1002	1.51	39.69	38.18
NP1	1000	5.51	3.61	0.1142	1.65	30.93	29.28
NP2	1000	4.89	2.95	0.0934	1.83	28.33	26.49
NP3	1000	5.24	2.94	0.0930	1.09	35.63	34.53
NP4	1000	5.75	3.30	0.1045	1.35	29.75	28.41
NP5	1000	4.79	3.32	0.1051	1.36	41.00	39.64

**Table 4 nanomaterials-11-02197-t004:** Estimate of the average diameter of NP0.

NP0	Diameter (nm)	SD (nm)
Raman	7.8	0.3
TEM	5.4	3.2
SMM	4.4	2.9

**Table 5 nanomaterials-11-02197-t005:** Data from measurements of NPs.

Samples	NP0	NP1	NP2	NP3	NP4	NP5
Magnetization at 0.5 T (Am^2^/kg)	25.9	24.4	21.6	23.9	25.7	25.1
Magnetization at 2.0 T (Am^2^/kg)	32.1	30.2	27.8	29.5	32.2	31.8

## Data Availability

The data presented in this study are available on request from the corresponding author.

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
