# Peer review of "Magnetic Characterization by Scanning Microscopy of Functionalized Iron Oxide Nanoparticles"

_nanomaterials, 2021, doi:10.3390/nano11092197_

Round 1

Reviewer 1 Report

Authors presented very interesting approach for characterization of magnetic nanoparticles. Especially important is comparison of TEM, Raman and SMM.

I have only few suggestions before publishing.

Could you please provide sketch of the SMM?

Line 145: please, provide references for the Raman modes.

At the last paragraph of introduction, you describe the techniques that will be used. Please, mention the importance of VSM as well and support this statement by ref. DOI: 10.3390/NANO10101990

Author Response

Dear Editor and reviewers, we would like to thank you for the valuable review of our paper.

Reviewer 1:

Authors presented very interesting approach for characterization of magnetic nanoparticles. Especially important is comparison of TEM, Raman and SMM.

Dear Reviewer 1, we would like to thank you for the valuable review of our paper.

I have only few suggestions before publishing.

Could you please provide sketch of the SMM?

Response:

A sketch of the SMM equipment can be found on references 33 (doi.org/10.1016/j.jmmm.2019.166300) and 34 (doi.org/10.3390/s19071636) and a sentence was added to the first paragraph  of section 3.3 citing those articles, to make this information clearer.

Lines: 256-257:

“A drawing sketch of the microscope assembly can be found on references [33-34].”

Line 145: please, provide references for the Raman modes.

Response:

The following references have been added:

  • Iconaru, S. L., Guégan, R., Popa, C. L., Motelica-Heino, M., Ciobanu, C. S., & Predoi, D. (2016). Magnetite (Fe3O4) nanoparticles as adsorbents for As and Cu removal. Applied Clay Science, 134, 128–135. doi:10.1016/j.clay.2016.08.019
  • Torres-Gómez, N., Nava, O., Argueta-Figueroa, L., García-Contreras, R., Baeza-Barrera, A., & Vilchis-Nestor, A. R. (2019). Shape Tuning of Magnetite Nanoparticles Obtained by Hydrothermal Synthesis: Effect of Temperature. Journal of Nanomaterials, 2019, 1–15. doi:10.1155/2019/7921273
  • Benhammada, A., Trache, D., Kesraoui, M., & Chelouche, S. (2020). Hydrothermal Synthesis of Hematite Nanoparticles Decorated on Carbon Mesospheres and Their Synergetic Action on the Thermal Decomposition of Nitrocellulose. Nanomaterials, 10(5), 968. doi:10.3390/nano10050968
  • Benhammada, A., Trache, D., Kesraoui, M., & Chelouche, S. (2020). Hydrothermal Synthesis of Hematite Nanoparticles Decorated on Carbon Mesospheres and Their Synergetic Action on the Thermal Decomposition of Nitrocellulose. Nanomaterials, 10(5), 968. doi:10.3390/nano10050968
  • Fouad, D. E., Zhang, C., Mekuria, T. D., Bi, C., Zaidi, A. A., & Shah, A. H. (2019). Effects of Sono-assisted modified precipitation on the crystallinity, size, morphology, and catalytic applications of hematite (α-Fe2O3) nanoparticles: A comparative study. Ultrasonics Sonochemistry, 104713. doi:10.1016/j.ultsonch.2019.104

At the last paragraph of introduction, you describe the techniques that will be used. Please, mention the importance of VSM as well and support this statement by ref. DOI: 10.3390/NANO10101990

Response:

The VSM technique was mentioned in the manuscript and the suggested reference was added, by the following sentence in the last paragraph of the introduction:

Lines: 92 - 98.

Parameters extracted from the SMM data can be used to build a MxH hysteresis curve with the respective magnetization values for each applied field. Those values were compared to vibrating sample magnetometer (VSM) measurements, at room temperature, where the sample oscillates close to a detection coil, and an induced voltage appears, corresponding to the magnetization of the sample and a superconducting magnet generates the external magnetic field [ref.36].

Reviewer 2 Report

Reviewed paper “Magnetic characterization by scanning microscopy of functionalized iron oxide nanoparticles” deals with an interesting topic and can be interesting for readers of Nanomaterials journal. This manuscript seems to provide interesting insights on a topic of magnetic nanomaterials, so it may be of interest to the magnetic system community.

Authors presents investigation on magnetic properties of magnetite (Fe3O4) nanoparticles functionalized with different Pluronic F-127 surfactant concentrations (Fe3O4@Pluronic F-127). Materials were obtained with using magnetic characterization method based on three-dimensional magnetic maps generated by scanning magnetic microscopy. Nanoparticles were prepared by a wet chemical reaction. These materials have a superparamagnetic behavior. Authors estimated that the nanoparticles' magnetic core diameter was about 5 nm. They prepared measurements by transmission electron microscopy imaging and diffraction with Raman spectroscopy.

This paper suits the requirements of the journal. The paper contains 16 figures, 5 tables and 4 formulas – figures are legible and good quality.

English of the paper is rather good – in my opinion the language of the paper should be a little improved. I am asking for corrections by a native speaker.

Primary criteria:

  1. Does this manuscript offer original data and innovative insights?

Yes, the results presented appear to be original and potentially innovative.

  1. Does the subject matter manuscript have wide scientific interest and potential applicability?

The subject matter is relevant to the magnetic system community and is potentially applicable to other systems.

  1. Have the authors remembered to cite previous seminal papers on the subject?

The authors have cited 38 literature items – the numbers is a sufficient but not all items are significant on the subject.

  1. Is the Supplementary Material (Supporting Information) helpful, appropriate and error-free?

N/A

I find some mistakes for example:

  • Introduction chapter – in my opinion should be correct. Authors should include new information about topic of a paper. More information based on worldwide (global) study. The list of references should also be changed.
  • Production and functionalization of magnetic nanoparticles chapter – please describe all equipment used in the experiment – please insert model of equipment (manufacturer, city, country).
  • Production and functionalization of magnetic nanoparticles chapter should be named Experimental.
  • In my opinion Conclusions chapter is to short and should be a little changed. It should be slightly condensed – it should contain the most relevant information contained in Nanoparticles Characterization
  • References chapter is sufficient but papers cited in the references 23 from all 38 are older then 5 years – these publications constitute over 60 % of all cited papers. I propose to add some new (from the last 5 years). Author should include several modern papers of global research in this field.
  • In relation to the subject of the article, in particular to topics of magnetic characterization please see on articles of prof. A.K. Fedotov.
  • In the list of references I found 9 papers of the Authors of reviewed paper. Please indicate the differences in the studies presented in the article and their previous papers.
  • Figure captions - In the case of multi-component figures, the figure caption shall first give the name of the figure followed by references (a), (b) and so on – figures 4, 11, 15 and 16.
  • Author should use space between a numerical value and a unit – for example in the line 283 (it is 293K and should be 293 K). Please apply throughout your paper.
  • Please describe all the variables found in the formulas.
  • Please prepare a literature review according to the guidelines of the Nanomaterials

The results obtained are interesting and promising. The manuscript can be accepted for publication in Nanomaterials journal after MAJOR corrections.

Author Response

Reviewer 2:

Reviewed paper “Magnetic characterization by scanning microscopy of functionalized iron oxide nanoparticles” deals with an interesting topic and can be interesting for readers of Nanomaterials journal. This manuscript seems to provide interesting insights on a topic of magnetic nanomaterials, so it may be of interest to the magnetic system community.

Authors presents investigation on magnetic properties of magnetite (Fe3O4) nanoparticles functionalized with different Pluronic F-127 surfactant concentrations (Fe3O4@Pluronic F-127). Materials were obtained with using magnetic characterization method based on three-dimensional magnetic maps generated by scanning magnetic microscopy. Nanoparticles were prepared by a wet chemical reaction. These materials have a superparamagnetic behavior. Authors estimated that the nanoparticles' magnetic core diameter was about 5 nm. They prepared measurements by transmission electron microscopy imaging and diffraction with Raman spectroscopy.

Dear Reviewer 2, we would like to thank you for the valuable review of our paper.

This paper suits the requirements of the journal. The paper contains 16 figures, 5 tables and 4 formulas – figures are legible and good quality.

English of the paper is rather good – in my opinion the language of the paper should be a little improved. I am asking for corrections by a native speaker.

Primary criteria:

  1. Does this manuscript offer original data and innovative insights?

Yes, the results presented appear to be original and potentially innovative.

  1. Does the subject matter manuscript have wide scientific interest and potential applicability?

The subject matter is relevant to the magnetic system community and is potentially applicable to other systems.

  1. Have the authors remembered to cite previous seminal papers on the subject?

The authors have cited 38 literature items – the numbers is a sufficient but not all items are significant on the subject.

  1. Is the Supplementary Material (Supporting Information) helpful, appropriate and error-free?

N/A

I find some mistakes for example:

  • Introduction chapter – in my opinion should be correct. Authors should include new information about topic of a paper. More information based on worldwide (global) study. The list of references should also be changed.

Response:

The text has been revised and rewritten.

Lines: 46 - 56.

Lines: 78 - 81.

Lines: 92 - 98.

We also added new references:

Lines: 392 – 401.

Lines: 450 - 471.

Lines: 487 - 498.

Lines: 508 - 516.

  • Production and functionalization of magnetic nanoparticles chapter – please describe all equipment used in the experiment – please insert model of equipment (manufacturer, city, country).

Response:

The text has been revised and rewritten.

Lines: 112 - 113.

Line:  128.

Line:  136.

  • Production and functionalization of magnetic nanoparticles chapter should be named Experimental.

Response:

The text has been revised and rewritten.

Line: 104.

  • In my opinion Conclusionschapter is to short and should be a little changed. It should be slightly condensed – it should contain the most relevant information contained in Nanoparticles Characterization

Response:

The conclusion chapter was rewritten and more information about the magnetic characterization of the nanoparticles was added.

Lines: 360 - 362.

Lines: 363 - 364.

Lines: 368 - 382.

  • Referenceschapter is sufficient but papers cited in the references 23 from all 38 are older then 5 years – these publications constitute over 60 % of all cited papers. I propose to add some new (from the last 5 years). Author should include several modern papers of global research in this field.

Response:

We added new references:

The text has been revised and rewritten.

Lines: 392 – 401.

Lines: 450 - 471.

Lines: 487 - 498.

Lines: 508 - 516.

  • In relation to the subject of the article, in particular to topics of magnetic characterization please see on articles of prof. A.K. Fedotov.

Response:

The authors thank the suggestion, but no ‘A. K. Fedotov’ was found on this subject. Further research will be done and added to future works

  • In the list of references I found 9 papers of the Authors of reviewed paper. Please indicate the differences in the studies presented in the article and their previous papers.

Response:

We understood the reviewer's question and decided to remove from the paper 5 references to these references and replace it with more current references that are not from the authors themselves.

In relation to the references that were left, it dealt with the equipment used in the assembly or in the method of measuring or manufacturing the nanoparticles.

  • Figure captions - In the case of multi-component figures, the figure caption shall first give the name of the figure followed by references (a), (b) and so on – figures 4, 11, 15 and 16.

Response:

The figures captions were changed according to the referee’s suggestion

  • Author should use space between a numerical value and a unit – for example in the line 283 (it is 293K and should be 293 K). Please apply throughout your paper.

Response:

The manuscript was revised, and the units’ format were corrected

  • Please describe all the variables found in the formulas.

Response:

The variables are described in each equation. On equation 2, the ΔL variable was changed to L, as appears in equation 1.

Line: 183.

  • Please prepare a literature review according to the guidelines of the Nanomaterials

Response:

The references section was formatted accordingly

The results obtained are interesting and promising. The manuscript can be accepted for publication in Nanomaterials journal after MAJOR corrections.

Dear Reviewer 2, we would like to thank you for the valuable review of our paper.

Round 2

Reviewer 2 Report

In corrected paper Magnetic characterization by scanning microscopy of functionalized iron oxide nanoparticles” Author has properly addressed the concerns from the referee. All my remarks have been included in the revised document. Below you will find my comments on the attached answers.

Referring to my substantive reservations – the authors made the necessary modifications. They changed the text of the article and removed stylistic and grammatical errors.

Author reformatted and extended Introduction and Conclusion chapters. Author significantly reformat the entire article. They changed figure captions which were suggested. They modified a list of a references.

The manuscript can be accepted for publication in Nanomaterials journal in the current form.

Author Response

Dear Academic Editor, we would like to thank you for the valuable review of our paper.

We sincerely apologize, as there was a confusion when sending responses to the reviewers and to the editor.

Could not make technic measurements like Mossbauer or XMCD.

However, considering that the electron diffraction patterns do not allow to discriminate the magnetite (Fe3O4) of maghemite (γ-Fe2O3). But we use Raman spectroscopy, a technique commonly used in characterizing of different oxide polymorphs by present distinct Raman signatures, either by difference in crystal structure or by change in oxidation state, and among these oxides is iron oxide.

As mentioned in the abstract, the particles were kept at a controlled temperature of 20 °C and, since their synthesis, they have the same magnetization value, even after several months. Also, Raman spectroscopy results show that there was no oxidation of the samples during the 6 months, indicating no oxidation of the material. 

We believe we can explore are points with new measures in future work (Paper).

The paper with the new modifications is attached!

The text has been modified:

Line: 78.

Lines:  80-82.

Line: 148: Table 1

Pure Fe3O4 changed by: 0.000

Lines: 369-370.

"The as-prepared samples, stored at approximately 20 °C, showed no change in the Raman spectrum, even after six months"
